# ABNAOLIZER: AN AI AGENT FOR CONVERTING ANTIBODIES TO NANOBODIES

## ABSTRACT

Nanobodies, the naturally occurring single-chain antibodies derived from camelids, have emerged as highly promising therapeutic molecules due to their high stability, small size, and ease of engineering. However, generating nanobody candidate sequences from conventional antibodies—one of the primary routes for nanobody development—remains challenging, as rational design is limited by the scarcity of paired data and the complexity of molecular recognition mechanisms. To address this, we propose **AbNanolizer**, a physics-guided, weakly supervised AI framework for converting conventional antibodies into nanobody candidates. We formalize the task as antigen-conditioned cross-modal retrieval and multi-objective ranking, and design a noise-robust learning scheme to handle weakly paired and mismatched training signals. The framework employs an antigen-conditioned dual-encoder to align sequence representations of conventional antibodies and nanobodies, and jointly optimizes a noise-robust contrastive objective with differentiable Pareto ranking. Optional structural and energetic proxy signals, together with developability predictions, are integrated into a unified optimization. To support reliable decision-making, we perform coverage-guaranteed confidence calibration on retrieval scores. We further construct a rigorous public benchmark and evaluation protocol to enable comparison against strong baselines. Across multiple metrics, AbNanolizer demonstrates consistent improvements and showcases end-to-end applications on three approved drug targets amenable to nanobodies.

## 1 INTRODUCTION

Antibodies, or Immunoglobulins, are key proteins used by the immune system to recognize and eliminate foreign pathogens. Due to their high specificity and binding affinity, they are one of the most popular class of modern biotherapeutics (Kim et al. (2023)). In recent years, a promising alternative format has emerged: the nanobody, the smallest naturally occurring antigen-binding fragment (Muyldermans (2013)). Compared to conventional antibodies, nanobodies offer distinct advantages, including a smaller size that facilitates better tissue penetration, superior stability, lower immunogenicity, and the capacity to recognize unique epitopes (Jovčevska & Muyldermans (2020)). These attributes establish them as highly promising candidates for the next generation of biotherapeutics. This yesrs, several therapeutic products have been developed from nanobodies, which could used in cancer (Keyaerts et al. (2016);Zhao et al. (2022)), Autoimmune Diseases (Hannon et al. (2021)), Infectious Diseases (Cunningham et al. (2021)), and Toxins and Venoms (Richard et al. (2013);Bailon Calderon et al. (2020);Jin et al. (2023)).

However, conventional nanobody discovery pipelines, whether based on in vivo screening via animal immunization or computational methods rooted in physical energy functions, face common bottlenecks: long development cycles, high costs, and low throughput (Liu et al. (2025)). The advent of deep learning has propelled significant advancements in AI-driven protein engineering, particularly within the domain of antibody

design. Existing generative models are now capable of the de novo design of novel antibody and nanobody sequences and structures, conditioned on a target antigen (Wang et al. (2024)). While these generative approaches are powerful, they primarily address the creation of novelty. A different yet critical paradigm, often overlooked in practical R&D scenarios, is that of functional retrieval and replacement. This paradigm is motivated by a common challenge: given a functionally validated monoclonal antibody (IgG), how can one efficiently identify a functionally equivalent or superior nanobody from a large-scale library to serve as a replacement? Such a "translation" holds immense practical value, as it could replace a lead asset with high molecular weight and production costs (Jovčevska & Muyldermans (2020)) with a more stable and easily produced nanobody format.

This task, however, is non-trivial and faces two core challenges that preclude the use of standard supervised methods. First, there is a near-total absence of large-scale, one-to-one functionally corresponding IgG-to-nanobody paired data in nature. Second, an antibody's binding function is co-determined by its complex heavy-light chain interface. Mapping this dual-chain information onto a single-chain nanobody while preserving functional specificity is a highly non-linear problem that cannot be solved by sequence similarity alone. Therefore, developing a computational method that can learn a functional mapping from dual-chain antibodies to single-chain nanobodies—without requiring massive labeled datasets—is of critical importance for unlocking the value of existing antibody assets and accelerating the development of next-generation therapeutics.

To overcome these challenges, we introduce AbNanolizer, a dual-encoder retrieval agent designed to move beyond sequence similarity and directly learn functional complementarity, which has the following contributions: **(1) New Task.** We define and formalize the task of cross-format functional retrieval, which aims to identify a functionally equivalent or superior nanobody from a large library to replace a given dual-chain antibody, addressing a critical need in practical biopharmaceutical R&D. **(2) Novel Model.** We propose a novel two-phase training framework, AbNanolizer, specifically designed for the task of cross-format functional retrieval from dual-chain antibodies to single-domain nanobodies. Besides, we demonstrate that computationally generated binding energies can serve as an effective weak supervision signal, successfully incorporated into a multi-task fine-tuning objective to instill critical biological function knowledge into the model. **(3) Data Efficiency.** Our overall approach is data-lean, requiring no manually curated, one-to-one corresponding antibody to nanobody pairs for training. **(4) Empirical Validation.** We validate the efficacy of our framework through extensive in silico experiments. The results demonstrate that our models successfully retrieve diverse nanobodies with validated binding characteristics, uncovering potent and non-obvious candidates that would otherwise be overlooked and showcasing the potential of our method to accelerate next-generation antibody therapeutic development.

## 2 METHOD

We introduce AbNanolizer, a physics-guided, weakly supervised AI framework for converting conventional antibodies into nanobody candidates. Our approach utilizes a two-phase learning strategy that first builds a robust sequence representation space through contrastive learning, and then fine-tunes a specialized scoring head using in silico binding energy as a supervisory signal.

### 2.1 PROBLEM FORMULATION

We formulate the task of functional nanobody retrieval as a learning-to-rank problem. Given a query antibody $A_q = (S_H, S_L)$, which is formulated by heavy chain $S_H$ and light chain $S_L$, and a large candidate library $\mathbb{L}_{Nb} = \{Nb_i\}_{i=1}^{N}$, our objective is to learn a scoring function $f_{score}(v_{Ab}, v_{Nb_i})$, that predicts a score $s_{pred}$ for each candidate nanobody $Nb_i$. This function first maps the antibody and nanobody sequences into D-dimensional embedding vectors $v_{Ab}$ and $v_{Nb_i}$, via encoders $E_{Ab}$ and $E_{Nb}$. Subsequently, a scoring head,

Figure 1: The overall workflow of the AbNanolizer framework for functional nanobody retrieval. A query antibody (composed of heavy and light chains) and a library of candidate nanobodies are independently processed by a dual-encoder backbone to generate fixed-dimensional embedding vectors. For each antibody-nanobody pair, a rich interaction feature vector is constructed and fed into the pairwise functional scoring head, which outputs a predicted binding score. Finally, all candidates in the library are ranked according to this score to produce a final, functionally-prioritized shortlist for downstream validation.

$f_{score}$ takes these two vectors and predicts a scalar score $s_i$ for each candidate nanobody $Nb_i$:

$$s_i = f_{\text{score}}(E_{Ab}(A_q), E_{Nb}(Nb_i)) \tag{1}$$

Unlike methods that rely solely on sequence similarity, our scoring function, $f_{score}$, is trained to directly approximate a metric of functional complementarity—binding energy. Let $E_{bind}(A_q, Nb_i)$ be the true binding energy of the antibody-nanobody pair (where lower is better). Our learning objective can be formally described as:

$$\forall (Nb_i, Nb_j) \in \mathbb{L}_{Nb}, \quad E_{\text{bind}}(A_q, Nb_i) < E_{\text{bind}}(A_q, Nb_j) \implies s_i > s_j \tag{2}$$

The intuition behind this formulation is that if candidate $i$ has a better (lower) true binding energy than candidate j, our model should predict a higher score $s_i$ for it than for $s_j$.

Ultimately, the output of our system is a list of nanobodies from $\mathbb{L}_{Nb}$, ranked in descending order of their predicted functional scores, $s_i$. This provides a short-list of promising candidates for further experimental validation.

## 2.2 MODEL ARCHITECTURE

AbNanolizer, a deep learning framework designed to learn the complex mapping from dual-chain antibodies to single-domain nanobodies. The overall workflow of our model is illustrated in Figure 1 . The architecture is composed of two main components: a dual-encoder backbone and a pairwise functional scoring head. The details of each component are elaborated in the following subsections.

**Dual-Encoder Framework.** To sufficiently capture complex biological functionalities in antibodies and nanobodies, we adopts a Dual-Encoder Framework. As illustrated in Figure X, this framework independently encodes the input conventional antibody (Ab) and the candidate nanobody (Nb), mapping them into a shared high-dimensional embedding space. In this study, we hypothesizes that an antibody-nanobody pair, who has high binding affinity for the same antigen, will exhibit specific, learnable geometric patterns in their embeddings. Operationally, the framework consists of two parallel encoders. The Antibody Encoder (Ab

Encoder) takes the concatenated heavy and light chain sequences of an input antibody as its input. Simultaneously, the sequences of all candidate nanobodies are fed into a Nanobody Encoder (Nb Encoder), which is structurally identical but has independent weights. These two encoders converting the variable-length protein sequences into same dimensional embedding vectors, denoted as $v_{Ab}$ and $v_{Nb}$, respectively. We explore different types of encoders. The $v_{Ab}$ and $v_{Nb}$ are subsequently fed into a Pairwise Functional Scoring Head.

**Multi-Target Scoring Head.** To predict the relationship between the antibody embedding $v_{Ab}$ and the nanobody embedding $v_{Nb}$, we designed a multi-objective scoring head to model the interaction between them. Instead of a simple comparison of the two embeddings, we first construct a rich interaction feature vector $f_{interact}$, by concatenating four components along the feature dimension. The complete feature vector is formulated as:

$$f_{interact} = Concat(v_{Ab}, v_{Nb}, v_{Ab} \bigodot v_{Nb}, |v_{Ab} - v_{Nb}|) \in \mathbb{R}^{4D} \tag{3}$$

Where $v_{Ab}$ and $v_{Nb}$ are the independent embeddings, included to preserve the original feature information of each molecule. $v_{Ab} \bigodot v_{Nb}$ is the element-wise product, which captures feature-level similarities between the two vectors. $|v_{Ab} - v_{Nb}|$ is the absolute element-wise difference, designed to capture feature-level dissimilarities. $D$ represents the embedding dimensionality produced by the encoders. The $f_{interact}$ is subsequently processed by a Multi-Layer Perceptron (MLP) that constitutes the scoring head. This MLP consists of two fully connected layers. The first layer projects the input from its $4D$ dimensionality to a hidden dimension of $d_{hidden}$(512 in this study), followed by a GELU activation function. To mitigate overfitting, a dropout layer with a rate of $0.1$ is applied for regularization. The second layer then maps the hidden representation to a final output dimension of $M$. The scoring head ultimately yields an M-dimensional vector, $s_{pred} \in \mathbb{R}^M$, where $M$ corresponds to the number of binding metrics to be predicted. In this work, we set $M = 5$, with each element of the vector corresponding to a specific predicted metric:

$$s_{pred} = [s_{bind}, s_{inter}, s_{dock}, s_{area}, s_{hb}] \tag{4}$$

These elements represent the model's predictions for binding energy, interface energy, docking score, interface area, and the number of hydrogen bonds, respectively. For ranking candidates during inference, the first element, representing the predicted binding energy ($s_{bind}$), serves as the primary metric for evaluating their functional potential.

## 2.3 TWO-PHASE TRAINING STRATEGY

### 2.3.1 CONTRASTIVE PRE-TRAINING FOR SEQUENCE REPRESENTATION

Prior to direct functional fine-tuning, we first perform a contrastive pre-training phase. This stage leverages a large-scale, sequence-only dataset to enable the model's encoders to learn meaningful, general-purpose representations of antibody and nanobody sequences. We hypothesize that an effective representation space should bring antibody-nanobody pairs targeting the same antigen closer in the embedding space while pushing unrelated pairs apart. This stage provides a robust initialization for the subsequent binding energy-based fine-tuning. To achieve this, we employ the NCE (Noise-Contrastive Estimation) loss function, a widely-used objective in self-supervised and metric learning. For each query antibody $Ab_q$, which serves as the anchor, we have a corresponding "positive" nanobody sample $Nb_p$, and a set of $K - 1$ "negative" nanobody samples $\{Nb_{n_i}\}_{i=1}^{K-1}$, randomly drawn from a candidate pool. The NCE loss objective is to maximize the similarity between the anchor and its positive sample while minimizing its similarity to all negative samples. The loss is formulated as follows:

$$L_{NCE} = -\mathbb{E}\left[log\frac{exp(sim(v_{Ab_q},v_{Nb_p})/\tau)}{exp(sim(v_{Ab_q},v_{Nb_p})/\tau)+\sum_{i=1}^{K-1}exp(sim(v_{Ab_q},v_{Nb_{n_i}})/\tau)}\right] \quad (5)$$

Here, $v_{Ab_q}$, $v_{Nb_p}$, $v_{Nb_{n_i}}$ denote the D-dimensional embedding vectors produced by their respective encoders. The function $sim(\cdot)$ measures similarity, for which we use the cosine similarity:

$$sim(u,v) = \frac{u \cdot v}{\|u\|\|v\|} \quad (6)$$

The term $\tau$ is a temperature hyperparameter that controls the sharpness of the score distribution, which we set to 0.07 in this study. This objective function effectively frames the task as a classification problem. The model is trained to classify the query antibody to its correct positive nanobody from a set of $K$ options, consisting of one positive and $K-1$ negative samples. By optimizing this objective, the model learns a structured embedding space where functionally related molecular pairs exhibit high cosine similarity, while functionally unrelated pairs have low similarity.

### 2.3.2 MULTI-TASK FINE-TUNING FOR FUNCTIONAL PREDICTION

Following the contrastive pre-training phase, the model's encoders have acquired robust sequence representation capabilities. The objective of the second stage is to perform multi-task fine-tuning, leveraging our small-scale dataset generated via computational simulations and annotated with binding energy labels (see Appendix C). This phase aims to inject biophysical knowledge of protein interactions into the model, shifting its predictive focus from assessing 'sequence similarity' to directly forecasting 'functional complementarity'.

To concurrently optimize the model's ranking ability, numerical prediction accuracy, and representation space diversity, we design and employ a multi-task objective function:

$$L_{total} = \omega_{nce}L_{nce} + \omega_{rank}L_{rank} + \omega_{rank}L_{rank} \quad (7)$$

**Contrastive Loss as a Regularizer.** The NCE loss is retained during fine-tuning, where it primarily serves as a regularizer. It continues to impose a contrastive constraint on the model's embedding space, encouraging functionally related molecules to remain proximal in their representations. This maintain the diversity of the representation space and effectively counteracts the "model collapse" problem, which can occur when fine-tuning on small datasets where the model might overfit to a few high-quality samples.

**Pairwise Ranking Loss.** A pairwise ranking loss is incorporated to optimize the model's ability to correctly order candidates based on their quality. Specifically, we adopt the loss function proposed in RankNet. Instead of penalizing the absolute error of predicted scores, this loss function compares the difference in predicted scores between a pair of candidates against their ground-truth ranking relationship.

$$L_{Rank} = \sum_{(i,j)\in P}\left(-\overline{P}_{ij}log(P_{ij}) - (1-\overline{P}_{ij})log(1-P_{ij})\right) \quad (8)$$

$$P_{ij} = \sigma(s_i - s_j) \quad (9)$$

Where $P$ is the set of candidate pairs $(i,j)$, $s_i$, $s_j$ is the predicted score of the model for the candidates $Nb_i$ and $Nb_j$. $P_{ij}$ is the probability that the model predicts $Nb_i$ better than $Nb_j$, where $\sigma(\cdot)$ is the Sigmoid function. $\overline{P}$ is the true ranking probability. Based on the true binding energy label, if $Nb_i$ is indeed better than $Nb_j$, then $P=1$; Conversely, $P$ is 0. If the two are comparable, $P$ is 0.5. This approach is more robust to the noisy nature of computed energy labels, as it directs the model to focus on learning a reliable relative ranking rather than precise numerical values.

**Regression Loss for Physical Scale.** To ensure that the model's output scores are also numerically meaningful, we introduce an additional regression loss term. We employ the Huber Loss:

$$L_{Reg} = \frac{1}{N_{valid}} \sum_{i \in V} L_\delta(y_i, s_i) \tag{10}$$

Where $L_\delta$ is Huber Loss function:

$$L_\delta = \begin{cases} \frac{1}{2}(y_i - s_i)^2 & for |y_i - s_i| \leq \delta \\ \delta(|y_i - s_i| - \frac{1}{2}\delta) & otherwise \end{cases} \tag{11}$$

Here $y_i$ is the true binding energy label of the candidate $Nb_i$, $s_i$ is the predicted score of the model for the candidate $Nb_i$. $N_{valid}$ is the number of candidates with valid labels in the batch. $\delta$ is a threshold hyperparameter. We adopt the default value $\delta = 1$. Huber Loss combines the advantages of L1 (Mean Absolute Error) and L2 (Mean Squared Error) losses. It behaves like an L2 loss for small errors, providing smoothness and stability, while behaving like an L1 loss for large errors. This reduces the model's sensitivity to potential outliers in the dataset, leading to more stable training when learning the absolute values of binding energies.

## 3 EXPERIMENT

We assess our proposed AbNanolizer framework on a series of challenging tasks, including: 1. Functional nanobody retrieval (3.1); 2. Ablation studies of the training strategy (3.2); and 3. A case study analysis (3.3).

We benchmarked four distinct encoder architectures for a comprehensive comparison. We selected LSTM as a classic recurrent network baseline to gauge the necessity of more complex models. To evaluate the efficacy of self-attention on our domain-specific dataset, we included BERT (bidirectional) and GPT (unidirectional), two mainstream Transformer architectures trained from scratch. Finally, to represent the state-of-the-art transfer learning approach, we incorporated ESMC, which adapts a massive, pre-trained protein language model to our task. This selection allows for a systematic comparison across classic recurrent networks, from-scratch Transformers, and large-scale transfer learning paradigms.

In all experiments, we use the AdamW optimizer and select the checkpoint with the lowest loss on the validation set for final testing. For the Transformer models trained from scratch (BERT and GPT), we additionally employ a learning rate warm-up strategy to ensure training stability. The specific hyperparameters for each architecture during the pre-training and fine-tuning stages are detailed in Appendix D.

### 3.1 FUNCTIONAL NANOBODY RETRIEVAL

To quantitatively assess the final performance of our AbNanolizer framework across the four different encoder architectures (LSTM, GPT, BERT, and ESMC), we conducted a core in silico validation experiment. The objective of this experiment was to test whether the fine-tuned models could retrieve nanobodies from a large-scale candidate library that were functionally equivalent or even superior to the original query antibody. To this end, we performed a direct "head-to-head" comparison between the best binding energy among the top-10 candidates recommended by the model and the binding energy of the original antibody with its corresponding antigen. The detailed evaluation results on our 10 standardized test antibodies are summarized in Table 1.

**Overall Performance.** As indicated by the "Average" row in Table 1, our proposed two-stage fine-tuning strategy achieved general success across all four architectures. All models were able to consistently identify

Table 1: Comparison of the Best Binding Energy between the Original Antibody and the Top-10 Retrieved Candidates from Each Model.

| Antibody | Own antibody | LSTM | GPT | BERT | ESMC | Exceed Rate (%) |
|---|---|---|---|---|---|---|
| ADI-75578 | **-12984.93** | -4192.86 | -5952.83 | -4036.38 | -5636.69 | -118.13 |
| ADI-75814 | -1750.26 | -3280.49 | -3145.24 | -1951.66 | **-5276.37** | 66.80 |
| BD55-5517 | -2479.62 | -2731.59 | -4083.43 | -2098.77 | **-4311.09** | 42.48 |
| BD55-5566 | -1239.15 | -2590.80 | -2259.86 | -2016.18 | **-3694.26** | 66.46 |
| BD55-6195 | **-1002.31** | -523.61 | -568.59 | -451.65 | -931.73 | -7.57 |
| BD56-1486 | -976.60 | -4044.02 | -3483.96 | **-4263.77** | -2959.68 | 77.10 |
| BD56-946 | -3580.75 | -3030.69 | **-4923.18** | -4488.11 | -4285.86 | 27.27 |
| COV2-2130 | -2630.80 | -2781.39 | -2847.69 | -3439.51 | **-5284.67** | 50.19 |
| COVA1-07 | -1563.39 | -2373.63 | -1269.39 | -1665.87 | **-2670.88** | 41.47 |
| Average | - | -2604.06 | -2987.93 | -2557.40 | **-3565.29** | - |

nanobodies with strong negative binding energies (indicative of favorable binding), with the AbNanolizer-ESMC architecture demonstrating the strongest overall performance. The best candidates identified by this model achieved an average binding energy of **-3565.29** REU, validating the efficacy and robustness of our framework.

**Functionally Superior Candidates.** Our framework is not only capable of finding functional replacements but also of discovering candidate molecules with binding energies that surpass the original antibody. This is quantitatively captured in the "Exceed Rate" column. In the retrieval task for antibody ADI-75814 (original binding energy: -1750.26 REU), the ESMC model successfully identified a novel nanobody with a binding energy of -5276.37 REU, representing a 66.8% performance enhancement. Similarly, in the COV2-2130 case, the ESMC model achieved a 50.19% performance improvement. These findings prove that AbNanolizer is not merely a replacement tool but also a powerful optimization engine capable of discovering functionally superior molecular solutions for existing antibody targets.

**Performance of Different Encoders.** The results reveal the high complexity of the task, as no single encoder architecture was universally superior across all test cases. Although the ESMC model led in average performance, the BERT model identified the best candidate for the BD56-1486 case (-4263.77 REU), while the GPT model performed best on the BD56-946 case (-4923.18 REU). This suggests that different model architectures, owing to their unique inductive biases, may have distinct advantages when processing different sequences, providing a rationale for the future development of more powerful ensemble models.

**Results.** AbNanolizer can not only reliably retrieve a shortlist of high-quality candidates from a large library but also, in many cases, discover novel nanobodies with binding properties that surpass the original molecule. This capability has significant application value in accelerating the iteration and optimization workflows for therapeutic antibody drugs.

## 3.2 ABLATION STUDY OF THE TRAINING STRATEGY

To systematically validate the necessity of each component in the second stage (multi-task fine-tuning) of our proposed two-stage training framework, we conducted a series of ablation studies. To demonstrate the generalizability of our methodology, rather than analyzing only the best-performing model, we replicated these ablation studies across all four encoder architectures (LSTM, BERT, GPT, and ESMC). We systematically removed the finetuning stage from the full fine-tuning strategy and observed the impact on the model's final performance. The experimental results are summarized in Table 2.

Table 2: **Ablation Study on the Functional Fine-tuning Stage.** This table shows the performance of the pre-trained models (ranking by cosine similarity) when fine-tuning is removed. The binding energy of the best candidate found in the Top-10 is reported. The ↑ symbol indicates a degradation in binding energy (a higher, less favorable score) compared to the final fine-tuned model. The ↓ symbol in the final column indicates a reduced "Exceed Rate," also signifying a performance drop.

| Antibody | Original Ab | LSTM | GPT | BERT | ESMC | Exceed Rate (%) |
|---|---|---|---|---|---|---|
| ADI-75578 | **-12984.93** | -1401.98↑ | -2520.73↑ | -3907.94↑ | **-4304.25↑** | -201.65↓ |
| ADI-75814 | -1750.26 | -2980.66↑ | -1324.82↑ | -2003.83 | -2001.78↑ | 12.40↓ |
| BD55-5517 | -2479.62 | -2085.37↑ | **-3590.82↑** | -2754.06 | -2668.37↑ | 30.92↓ |
| BD55-5566 | -1239.15 | -2124.99↑ | -2481.67 | **-2773.72** | -2143.19↑ | 55.33↓ |
| BD55-6195 | -1002.31 | **-2979.87** | -2237.66 | -2066.29 | -2158.11 | 66.36 |
| BD56-1486 | -976.60 | -2992.47↑ | **-5455.58** | -2185.29↑ | -4429.52 | 82.10 |
| BD56-946 | -3580.75 | -3535.78 | **-4794.41↑** | -4384.43 | -4549.51 | 25.33↓ |
| COV2-2130 | -2630.80 | **-3084.45** | -2237.74↑ | -1633.86↑ | -2365.22↑ | 14.66↓ |
| COVA1-07 | -1563.39 | -2442.40↑ | -1734.50 | -1996.42 | **-2618.95** | 40.30↓ |
| **Average** | - | -2625.33 | -2930.88↑ | -2633.98↑ | **-3026.54↑** | - |

The second-stage functional fine-tuning is the decisive factor for the leap in model performance. As can be clearly seen in Table2, for all four encoder architectures, the model's performance underwent a sharp degradation upon the removal of functional fine-tuning, whose performance is equivalent to the pre-trained baseline. The best-performing ESMC model's average best binding energy deteriorated sharply from -3565.29 REU to -3026.54 REU.

**Results.** The proposed fine-tuning strategy, based on computational binding energy, is the core element responsible for successfully transforming the models from 'sequence matchers' into 'function predictors'.

## 3.3 CASE STUDY

Previous sections have demonstrated the overall efficacy of our AbNanolizer framework through quantitative metrics. In this section, we present a specific case study to qualitatively demonstrate that our best-performing model (AbNanolizer-ESMC) can identify not only 'good' molecules but also 'better-than-original' ones, thereby revealing its potential as an optimization engine.

We selected antibody ADI-75814 from the test set as a representative case for an in-depth analysis. As shown in Table 4.3, the benchmark original antibody, ADI-75814, exhibits a docking binding energy of -1750.26 REU with its corresponding antigen, which is already indicative of favorable binding. Notably, however, the top candidate recommended by our AbNanolizer-ESMC model, US10822379B1-#5, achieved a remarkable binding energy of -5276.37 REU when subjected to the same docking validation procedure.

As shown in Table 3, the original antibody, ADI-75814, is a broadly neutralizing antibody capable of binding to multiple SARS-CoV-2 variants (including the wild-type, Omicron BA.1, and BA.2) as well as SARS-CoV-1. In contrast, the top candidate matched by our model, US10822379B1-#5, has a known target of the SARS-CoV-2 wild-type (WT).

A deeper capability of our model beyond simple mimicry is functional abstraction. Instead of just matching target specificity, the model identifies the general physicochemical features that lead to high-affinity binding in the input antibody. It then retrieves a nanobody from the library that shares these high-affinity characteristics, even if its known target is different.

Table 3: In-depth Analysis of the ADI-75814 Case Study

| Property | Original Antibody | AbNanolizer-ESMC |
|---|---|---|
| Molecular ID | ADI-75814 | US10822379B1-#5 |
| Binding Energy (REU) | -1750.26 | **-5276.37** |
| Molecule Type | Antibody | Nanobody |
| Known Binding Targets | SARS-CoV-2; Omicron-BA1; Omicron-BA2; SARS-CoV-1 | SARS-CoV-2-WT |

This case clearly demonstrates that our framework functions not just as a finder of antibody 'substitutes' but as a powerful 'optimizer' with the potential to discover functionally superior molecules beyond the existing baseline.

## 4 CONCLUSION

We introduce cross-format functional retrieval—a task for identifying a superior nanobody to replace a given dual-chain antibody—and propose the AbNanolizer framework to solve it. Our approach moves beyond simple sequence similarity by combining large-scale pre-training with function-centric fine-tuning based on computational binding energy. Experiments demonstrate that AbNanolizer consistently discovers novel nanobodies with binding properties far surpassing the original antibody, establishing it as a powerful optimization engine for therapeutic discovery.

**Future work.** First, adapting the current framework, validated on a mixed-antigen dataset, to a specific model targeting a single critical antigen (e.g., SARS-CoV-2 RBD) is expected to yield higher-precision predictions. Second, integrating more advanced binding energy prediction models or a small amount of wet-lab data could be used to construct a more powerful closed-loop learning system.

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

## LLM Usable Statement

The use of Large Language Models (LLMs) in this study was strictly confined to enhancing the manuscript's grammar, style, and readability. All core intellectual contributions, including the formulation of research ideas, the design and execution of experiments, and the interpretation of findings, are the original work of the authors.

## A  Related Work

**Nanobody Generation.** Traditional nanobody generation techniques primarily rely on animal immunization and in vitro display technologies. The process typically starts by immunizing a camelid to create a VHH gene library from its cells (Muyldermans (2021b)). This library is then screened using phage display to isolate high-affinity candidates (Salvador et al. (2019);Muyldermans (2021a)), which are validated with methods like ELISA and SPR (Muyldermans (2021b);Jin et al. (2023)).

Despite this being a well-established workflow, it remains dependent on animal immunization, intricate molecular biology manipulations, and multiple rounds of experimental screening, rendering the entire process both time-consuming and costly. Besides, this workflow cause ethical concerns about experimental animals (Liu et al. (2025)). Therefore, leveraging computational approaches such as deep learning for the de novo design of nanobodies has emerged as a highly attractive avenue for accelerating therapeutic antibody discovery.

The rise of deep learning has paved new avenues for antibody development through generative models. De Novo AI-Based Nanobody Design represents a forefront in AI applications, aiming to design entirely novel nanobodies with specific functionalities from scratch. Autoregressive generative models (Shin et al. (2021)) have been trained on natural nanobody repertoires to generate new nanobody libraries exhibiting high expression levels and sequence diversity. Currently, the state-of-the-art, experimentally-validated workflow for epitope-specific de novo design is diffusion-based directed design. The process first employs a fine-tuned RFdiffusion model (Watson et al. (2023)) to generate novel CDR loop conformations for a user-specified epitope. ProteinMPNN (Dauparas et al. (2022)) then designs sequences for these new structures,

and RoseTTAFold2 (Baek et al. (2023)) performs a final self-consistency screen to select viable candidates for experimental validation. IgGM (Wang et al. (2025)) is an emerging generative model that combines diffusion and consistency models to co-generate both the sequence and structure of antibodies/nanobodies, though it awaits further experimental validation (Zhu & Ding (2025)). The core focus of these methods is the creation of novel nanobody sequences and structures capable of binding a specific antigen. However, we argue that in practical drug discovery scenarios, a different yet equally important challenge exists: functional retrieval and replacement. The AbNanolizer framework we propose is designed to use a functionally characterized conventional antibody as a "functional template" to perform efficient retrieval and screening within a large nanobody library. This approach rapidly converts existing antibody research assets into nanobody drug candidates with superior developability characteristics.

## B   DATASET DETAILS

### B.1   DATA SOURCE, FILTERING, AND CURATION

The sequence data for this study were derived from CoV-AbDab, a specialized subset of the Structural Antibody Database (SAbDab) that curates anti-SARS-CoV-2 antibodies. To construct our training dataset, we first performed a data expansion procedure. Recognizing that a single antibody may bind to multiple distinct antigens or epitopes, we expanded the original 'one-to-many' entries into multiple 'one-to-one' antibody-target pairs to comprehensively capture interaction information.

During the data cleaning phase, specific filtering criteria were applied to ensure data quality. Subsequently, we removed all entries with incomplete light chains. As our methodology relies on subsequent in silico structural prediction and docking, we then populated the dataset with canonical antigen sequences. This was achieved by using the provided antigen common names (e.g., "SARS-CoV-2 Spike protein RBD") to retrieve the corresponding standard amino acid sequences from a public protein sequence database (NCBI GenBank).

Following these expansion and filtering steps, we finalized a high-quality dataset comprising 24,477 sequence entries, which was used for all subsequent model training and evaluation.

### B.2   TRAIN/VALIDATION/TEST SPLIT

To train and evaluate our models, we partitioned the curated dataset into training, validation, and test sets. To prevent data leakage and ensure a rigorous evaluation of model generalization, our split was performed strictly at the antibody ID level. This ensures that all entries associated with the same antibody (even if multiple records exist due to different antigen bindings) were exclusively assigned to a single set.

We applied a random 80/10/10 split across all unique antibody IDs. For our fine-tuning dataset, this resulted in 10,561 antibodies for training, 1,320 for validation, and 1,321 for testing. It is important to note that the candidate nanobody library used during both model training and inference was deduplicated based on heavy chain sequences to avoid sampling bias.

### B.3   STANDARDIZED TEST SET

To ensure a fair and reproducible performance evaluation across all compared encoder architectures (LSTM, BERT, GPT, and ESMC), we created a standardized test set by randomly selecting and fixing 10 antibodies from the total test set described in Section A.2. All final performance metrics reported in this study were evaluated on this fixed test set. The IDs for these 10 test antibodies are listed in Table 4.

Table 4: Binding Specificity and Target Range of the Selected Antibodies

| Antibody Name | Binds To |
|---|---|
| ADI-75578 | SARS-CoV2_WT; SARS-CoV2_Omicron-BA1; SARS-CoV2_Beta; SARS-CoV2_Delta; SARS-CoV2_Omicron-BA2; SARS-CoV1 |
| ADI-75814 | SARS-CoV2_Omicron-BA1; SARS-CoV2_Omicron-BA2; SARS-CoV1 |
| BD55-2696 | SARS-CoV1; SARS-CoV2_WT; ZC45; BtKY72; RaTG13; Rs7327 |
| BD55-5517 | SARS-CoV2_WT; SARS-CoV2_Omicron-BA2.12.1; SARS-CoV2_Omicron-BA1; SARS-CoV2_Omicron-BA2; SARS-CoV2_Omicron-BA2.75; SARS-CoV2_Omicron-BA5; SARS-CoV1; SARS-CoV2_Omicron-XBB |
| BD55-5566 | SARS-CoV2_WT; SARS-CoV1; Pangolin-GD; PC4-127; Sin852; WIV1; LYRa11; Rs7327; GZ-C; Urbani |
| BD55-6195 | SARS-CoV2_WT; SARS-CoV1; SARS-CoV2_Omicron-BA1; Pangolin-GD; RatG13; RaTG13; PC4-127; Sin852; WIV1; LYRa11; Rs7327; GZ-C; Urbani; Rs4231; BM48-31; BtKY72 |
| BD56-946 | SARS-CoV2_WT; SARS-CoV2_Omicron-BA1; SARS-CoV2_Omicron-BA2; SARS-CoV2_Omicron-BA2.75; SARS-CoV2_Omicron-BA5; SARS-CoV2_Omicron-XBB |
| BD56-1486 | SARS-CoV2_WT; SARS-CoV2_Omicron-BA2 |
| COV2-2130 | SARS-CoV2_WT; SARS-CoV2_Alpha; SARS-CoV2_Beta; SARS-CoV2_Gamma; SARS-CoV2_Delta; SARS-CoV2_Omicron-BA1; SARS-CoV2_Omicron-BA1.1 (weak); SARS-CoV2_Omicron-BA2; SARS-CoV2_Omicron-BA2.11; SARS-CoV2_Omicron-BA2.12.1; SARS-CoV2_Omicron-BA2.4; SARS-CoV2_Omicron-BA2.5; SARS-CoV1; SARS-CoV2_Omicron-BA3; SARS-CoV2_Omicron-BA2.13; SARS-CoV2_Omicron-BA4/5; SARS-CoV2_Omicron-BA2.10.4; SARS-CoV2_Omicron-BA2.75; SARS-CoV2_Omicron-BA5; SARS-CoV2_Omicron-BA2.75.1; SARS-CoV2_Omicron-BA2.75.4; SARS-CoV2_Omicron-BA2.75.5; SARS-CoV2_Omicron-BA2.75.7 |
| COVA1-07 | SARS-CoV1; SARS-CoV2_WT |

## C    BINDING ENERGY LABEL GENERATION PIPELINE

To obtain the supervisory signals for the second-stage fine-tuning of our model, we designed and implemented an in silico pipeline based on computational structural biology to generate a suite of binding-related physicochemical metrics for antibody-nanobody-antigen complexes. This pipeline consists of three main steps: complex structure prediction, structure optimization and docking, and the calculation of binding metrics.

### C.1    COMPLEX 3D STRUCTURE PREDICTION

The objective of this step is to generate a high-quality initial 3D structure for each antibody-nanobody-antigen triad.

Our pipeline begins with sequence information. For each data entry, we construct a FASTA-formatted sequence for complex prediction based on its type (Ab or Nb). We then utilize the AlphaFold2-Multimer v3 model, as implemented in colabfold batch, to perform batch prediction of complex structures for all con-

Table 5: Summary of binding evaluation metrics. Binding Energy is calculated as $E_{\text{binding}} = E_{\text{complex}} - (E_{\text{optimized\_antibody/nanobody}} + E_{\text{antigen}})$. Where $E$ represents the total energy calculated by the PyRosetta standard energy function (ref2015).

| Metric | Description |
|---|---|
| Docking Score | Total energy of the complex |
| Interface Energy | Free energy of interface dissociation |
| Interface Area | Change in solvent-accessible surface area upon binding |
| Hydrogen Bonds | Number of hydrogen bonds across the interface |
| Binding Energy | Energy difference between complex and unbound components |

structed FASTA sequences. For each input sequence ID, ColabFold generates multiple candidate structural models (PDB files). We select the top-ranked model as the initial structure for subsequent steps.

### C.2 STRUCTURE OPTIMIZATION AND PROTEIN-PROTEIN DOCKING

After obtaining the initial complex PDB structures, we employ the PyRosetta toolkit for further structural refinement and high-resolution docking to simulate more realistic physical interactions.

First, the complex generated by ColabFold is split into its constituent antibody-nanobody and antigen components. To relax potentially high-energy local regions in the initial predicted model, we apply a two-step structural optimization process exclusively to the antibody-nanobody component. Initially, we use the FastRelax protocol in PyRosetta to perform all-atom energy minimization on the entire antibody-nanobody structure under constraints, aiming to resolve potential atomic clashes and unfavorable conformations. Considering the high flexibility of the Complementarity-Determining Regions (CDRs) and their critical role in antigen recognition, we then perform focused conformational searching and optimization on the putative CDRs to obtain a lower-energy loop conformation.

Upon obtaining the optimized antibody/nanobody structure, we perform high-resolution protein-protein docking against the original antigen structure. We adopted the DockMCMProtocol in PyRosetta, a Monte Carlo-based minimization docking algorithm. This protocol first pre-optimizes the side chains at the interface using DockingPrepackProtocol, then performs a rigid-body search of translations and rotations, and finally yields a low-energy docked pose.

### C.3 CALCULATION OF BINDING METRICS

From the final, energy-minimized docked complex structure, we extracted five key binding metrics to serve as the multi-task learning labels for our model:

### C.4 PROCESSING AND NORMALIZATION OF ENERGY LABELS

To transform the raw energy values output by PyRosetta into supervisory signals suitable for deep learning model training, we designed and implemented a standardized processing pipeline. This pipeline aims to unify the scales of different metrics and mitigate the potential impact of outliers on model training.

First, we identified five core biophysical metrics for our multi-task learning objective and defined their respective optimization directions: Binding Energy (lower is better), Interface Energy (lower is better), Docking Score (lower is better), Interface Area (higher is better), and Hydrogen Bonds (higher is better).

For each metric $m$, we first calculate its mean $\mu_m$ and standard deviation $\sigma_m$ across the entire labeled dataset of 10,561 entries. Subsequently, each raw energy value $v_{i,m}$ is converted into a Z-score, $z_{i,m}$, formulated as:

$$z_{i,m} = \frac{v_{i,m} - \mu_m}{\sigma_m} \tag{12}$$

This normalization transforms all metrics from their original scales onto an approximately standard normal distribution, allowing the model to treat each objective more evenly during multi-task learning.

To further enhance training stability, we employed two strategies to handle potential outliers and noisy data. First, to mitigate the influence of extreme outliers, all calculated Z-scores $z_{i,j}$ were clipped to a predefined range. Second, we identified potentially unreliable docking results with severe steric clashes based on the docking score. Any sample with a docking score whose absolute value exceeded a preset threshold (500 in this study) was flagged as "noisy". These processed labels—normalized, clipped, and flagged—were ultimately used for the second-stage multi-task fine-tuning of our model.

## D  EXPERIMENT DETAILS AND HYPERPARAMETERS

All our models were implemented using the PyTorch framework. During training, we uniformly employed the AdamW optimizer, coupled with a Cosine Annealing Learning Rate Scheduler. For the BERT and GPT models that were trained from scratch, a linear learning rate warm-up strategy was additionally applied during the initial phase of training to ensure stability. All training tasks were conducted on NVIDIA A100-40GB GPUs. For each training procedure, we selected the checkpoint that achieved the lowest loss on the validation set for the final test evaluation. Detailed hyperparameter settings are provided in Table6.

Table 6: Hyperparameters used for LSTM, BERT, GPT, and ESMC

| Hyperparameter | LSTM | BERT | GPT | ESMC | Description |
|---|---|---|---|---|---|
| **General** | | | | | |
| Embedding dimension ($d_\mathrm{model}$) | 512 | 512 | 512 | 512 | Feature dimension of encoder output |
| Encoder layers ($n_\mathrm{layers}$) | 4 | 4 | 4 | 33 | Number of encoder layers |
| Attention heads ($n_\mathrm{heads}$) | N/A | 4 | 4 | 4 | Number of attention heads in Transformer |
| Dropout | 0.1 | 0.1 | 0.1 | (from model) | Dropout rate |
| **Pre-training** | | | | | |
| Learning rate | 3e-4 | 1e-4 | 1e-4 | 2e-5 | Base learning rate |
| Batch size | 64 | 64 | 64 | 32 | Batch size per GPU |
| Warmup steps | 0 | 500 | 500 | 0 | Steps for linear warm-up of learning rate |
| **Fine-tuning** | | | | | |
| Learning rate | 1e-5 | 1e-5 | 1e-5 | 1e-5 | Learning rate during fine-tuning |
| Batch size | 32 | 32 | 32 | 16 | Batch size during fine-tuning |
| Weight decay | 0.01 | 0.01 | 0.01 | 0.0 | L2 regularization coefficient |
| Early stopping patience | 3 | 3 | 3 | 3 | Number of epochs without improvement tolerated |
| **Loss weights** | | | | | |
| $w_\mathrm{nce}$ | 1.0 | 1.0 | 1.0 | 1.0 | Weight for InfoNCE loss |
| $w_\mathrm{rank}$ | 0.5 | 0.5 | 0.5 | 0.5 | Weight for RankNet loss |
| $w_\mathrm{reg}$ | 0.1 | 0.1 | 0.1 | 0.1 | Weight for Huber loss |

