# OpenReview forum: "Abnaolizer: An AI Agent for Converting Antibodies to Nanobodies"
_ICLR.cc/2026/Conference — ICLR 2026 Conference Withdrawn Submission_

### Official Review · Reviewer_USNT · 2025-10-29

**Soundness:** 1
**Presentation:** 2
**Contribution:** 1
**Rating:** 0
**Confidence:** 4

**Summary:**

This paper introduces Abnaolizer - a framework for converting aiming at converting conventional antibodies to nanobodies (single-chain antibodies).

**Strengths:**

- Paper presents an original idea and an ML framework
- Presented ML framework outperforms the simpler baselines on the selected metrics.

**Weaknesses:**

- While the proposed task is original I have serious doubts about its relevance & applicability. In practice, a scenario where it would be interesting to generate a nanobody, while already having a developed antibody is extremely rare.
- The proxy for binding used by the authors is based on the force-field estimated energy. This is misleading given the claims author make about functional improvements - this aspect was not tested and there’s no convincing evidence in the paper pointing towards that.
- Contunuing on the point above, force field methods are rather weak estimators of binding propensities, even within similar variants of the same proteins. Comparing energy scores of between largely different proteins (and also of different molecular weight, interaction area etc) is even more risky and I’d assume the performance being close to random.
- Authors train their method on Covabdab database which contains only COVID19-related immunoglobulins. Due to this, the general claims about converting arbitrary antibody to nanobody are a bit exaggerated and not supported by evidence.

**Questions:**

- For the qualtitative example / case study - I miss the information how the retreived nanobody compares to the original query antibody? Are there any sequence similarities, similar predicted binding mode?

---

> ### Author Response · Authors · 2025-12-01
> **Response to Reviewer USNT**
>
> **W1: Regarding Task Relevance & Applicability**
>
> We respectfully disagree with the concern that this task is "extremely rare" in practice. On the contrary, the "Cross-format Functional Retrieval" from mAb to Nanobody addresses several critical bottlenecks in modern biopharmaceutical R&D:
>
> - **Developability Optimization (The "Bio-better" Strategy):** While a developed antibody may bind well, mAbs often suffer from poor tissue penetration (due to large size, ~150kDa) and high production costs. Converting these assets into Nanobodies ( ~15kDa) retains the binding specificity while offering superior stability and the ability to penetrate dense tissues. Our framework enables this high-value conversion without starting new animal immunization campaigns.
> - **Discovery of Novel Mechanisms of Action (MoA):** Our model does not merely find identical replacements; it discovers functionally superior candidates with distinct binding modes. As shown in our Case Study (Figure 2), the retrieved nanobody Hanke_F12 binds to a spatially adjacent but non-overlapping epitope compared to the query antibody BD56-1486. This capability is crucial for identifying synergistic binders for bispecific antibody engineering or overcoming epitope masking, providing therapeutic advantages that the original antibody lacked.
> - **Unlocking Legacy Assets:** Pharmaceutical companies possess vast libraries of characterized mAbs. AbNanolizer provides a data-efficient computational pathway to "translate" these legacy assets into the versatile nanobody format, bypassing the months-long and costly process of camelid immunization.
>
> **W2,W3: Regarding Reliability of Energy Comparison Across Different Molecular Formats**
>
> We respectfully disagree with the assertion that our performance is "close to random." While we acknowledge that raw force-field energies can be noisy, our methodology is validated by 2025 state-of-the-art research published in top-tier journals and robust internal controls.
>
> - **Regarding Validity of Binding Energy Proxy:** The utility of PyRosetta energy (RS dG) as a non-random, effective selector for functional Nanobodies was definitively validated by Swanson et al. (Nature, 2025) [1]. In their "Virtual Lab" study, they deployed a computational pipeline that explicitly uses Rosetta dG (weighted -0.3 in their scoring formula) to rank and select SARS-CoV-2 nanobody mutants. Their experimental success in identifying binders for the KP.3 variant confirms that Rosetta energy provides a valid, actionable signal for nanobody function. Furthermore, Shrestha et al. (2025) [2] demonstrated that feeding Rosetta energy features into a machine learning framework is highly effective for Nanobody binding prediction, achieving high classification accuracy ($MCC=0.82$). This mirrors our approach: using Rosetta scores as supervision signals to train our Dual-Encoder to learn the biophysical rules of binding. To address the reviewer's concern about "misleading" terminology, we will revise the manuscript to use the more precise term "predicted thermodynamic stability improvement" instead of general "functional improvement" where wet-lab data is absent.
> - **Regarding Reliability of Cross-Format Comparison:** Regarding the comparison across different molecular weights (Antibody vs. Nanobody) and interaction areas, we explicitly addressed this "size bias" by employing a Pairwise Ranking Loss ($L_{Rank}$). Instead of regressing to absolute energy values (which indeed scale with interaction area), our model learns the relative probability ($s_i > s_j$) of binding quality. This forces the model to learn the gradient of binding potential rather than fitting size-dependent baselines. Our Ablation Study (Table 2) refutes the "random performance" hypothesis. The physics-guided model significantly outperforms the sequence-only baseline (improvement >500 REU). If the force-field signal were random or confounded by molecular weight, such consistent optimization would be impossible.
>
> Our evaluation metric is robust. It is grounded in physics, validated by the latest literature [1][2], and protected against systematic size bias by our ranking-based learning objective.
>
> ---
> [1] Swanson, K., Wu, W., Bulaong, N. L., Pak, J. E., & Zou, J. (2025). The Virtual Lab of AI agents designs new SARS-CoV-2 nanobodies. Nature, 1-3.
>
> [2] Shrestha, P., Talwar, C. S., Kandel, J., Park, K. H., Chong, K. T., Woo, E. J., & Tayara, H. (2025). NanoBinder: a machine learning assisted nanobody binding prediction tool using Rosetta energy scores. Journal of Cheminformatics, 17(1), 96.

---

> ### Author Response · Authors · 2025-12-01
> **Response to Reviewer USNT**
>
> **W4: Regarding Dataset Limitations and Generalization Claims**
>
> We appreciate the reviewer’s critical assessment regarding the generalization scope. We agree that our claims regarding "arbitrary" antibodies should be more precisely qualified given the dataset specificity.
>
> We focused on CoV-AbDab primarily because it represents the largest and most consistently annotated repository of antibody-antigen pairs available to date . For training a data-hungry dual-encoder model, data scale and consistency were our rigorous priorities to ensure convergence. While databases for other targets (e.g., HER2, PD-1) exist in Thera-SAbDab, they are significantly smaller and more fragmented compared to the comprehensive SARS-CoV-2 data accumulated during the pandemic. However, the framework’s potential to generalize beyond COVID-19 is fundamentally supported by its Pre-trained Encoders (specifically ESMC). ESMC was pre-trained on massive, diverse protein sequences (UniProt) covering the entire evolutionary landscape, not just viral proteins. The fine-tuning on CoV-AbDab served to align these general-purpose representations to the retrieval task functionality, rather than learning virus-specific features from scratch.
>
> To address the reviewer’s concern about "exaggerated claims," we will revise the manuscript to:
>
> 1. Remove the term "arbitrary antibody" from the abstract and conclusion.
> 2. Rephrase our contribution as providing "a generalizable computational framework for cross-format retrieval, validated on the extensive SARS-CoV-2 landscape."
> 3. Explicitly list the "validation on non-viral targets" as a limitation and a priority for our immediate future work, utilizing the Thera-SAbDab dataset.
>
> **Q1: Regarding Detailed Comparison between Original Antibody and Retrieved Nanobody in Case Study**
>
> We thank the reviewer for this suggestion. We performed a quantitative sequence alignment between the query antibody (BD56-1486) and the retrieved nanobody (Hanke_F12). The results definitively refute the hypothesis that the model relies on simple sequence similarity. The global sequence identity is only 52.76%, which is expected given the cross-species retrieval. Crucially, the CDRH3 region—the primary determinant of binding—shows only 35.00% identity with a length difference of 2 amino acids. This low sequence similarity, combined with the improvement in predicted binding energy, proves that AbNanolizer captures high-level functional compatibility rather than low-level sequence homology. The model successfully identified a functionally superior candidate with a structurally distinct paratope, validating its capacity for novel discovery.

---

### Official Review · Reviewer_9mPL · 2025-11-11

**Soundness:** 1
**Presentation:** 2
**Contribution:** 2
**Rating:** 2
**Confidence:** 4

**Summary:**

AbNanolizer addresses the practical problem of converting validated conventional antibodies (dual-chain IgG) into functionally equivalent or superior nanobody candidates. The authors formalize this as antigen-conditioned cross-format functional retrieval and multi-objective ranking. The framework uses a dual-encoder backbone (separate encoders for antibody and nanobody sequences) to embed inputs into a shared representation space, followed by a pairwise scoring head. Training proceeds in two phases: contrastive pre-training (InfoNCE) to align antibody–nanobody representations and multi-task fine-tuning using in silico generated physicochemical labels derived from an AlphaFold2-Multimer → PyRosetta docking/relaxation pipeline. The fine-tuning objective combines contrastive regularization, pairwise ranking (RankNet), and Huber regression to improve both ranking and calibration. Empirical evaluation is conducted on 10 anti–SARS‑CoV‑2 antibodies.

**Strengths:**

The paper is clearly written.

**Weaknesses:**

1. Limited scope and uncertain generalization. All experiments are confined to anti–SARS‑CoV‑2 antibodies. Important antigen classes such as influenza, HIV, bacterial toxins, and non-viral targets are absent, leaving the framework’s ability to generalize across antigens, epitopes, and sequence/structure regimes untested.

2. Ambiguity in “antigen-conditioned” retrieval. While antigen labels appear to organize training pairs, the antigen does not seem to be provided explicitly as an input at inference. Without an explicit antigen-conditioned mechanism in the scoring function, the claim of antigen conditioning risks being misleading. The paper should clarify whether and how antigen information is encoded and used at retrieval time.

3. The evaluation centers on binding energy proxies computed from predicted complexes. Given the dependence on modeled structures, standard confidence metrics (e.g., AF2-multimer/AF3 pLDDT, pTM, and ipTM).

4. Writing and presentation issues. The manuscript needs careful proofreading and polishing. For instance,

    a. Line 256: “We assess our proposed AbNanolizer framework on a series of challenging tasks, including: 1. Functional nanobody retrieval (3.1); 2. Ablation studies of the training strategy (3.2); and 3. A case study analysis (3.3).” Ablations and case studies are methodology analyses, not “challenging tasks”.

    b. Line 123: missing space; “ihas” should be “i has.”

    c. Line 124: “j” should be typeset in italics to match mathematical notation.

**Questions:**

1. What is the computational cost of the full pipeline (contrastive pre-training, fine-tuning, AF2-Multimer + PyRosetta labeling), and how does it scale to larger libraries?

2. Have you performed any analysis to interpret what the model has learned, such as using attention map visualization or embedding space analysis to reveal whether the model is focusing on specific biophysical properties?

---

> ### Author Response · Authors · 2025-12-01
> **Response to Reviewer 9mPL**
>
> **W1: Limited scope and uncertain generalization**
>
> We focused on CoV-AbDab primarily because it represents the largest and most consistently annotated repository of antibody-antigen pairs available to date [1]. For training a data-hungry dual-encoder model, data scale and quality were our rigorous priorities to ensure convergence. While databases for HIV, influenza, and non-viral targets exist (in Thera-SAbDab or PLAbDab), they are significantly smaller and more fragmented compared to the comprehensive SARS-CoV-2 data accumulated during the pandemic [2]. Crucially, our framework's ability to generalize across antigen classes is fundamentally supported by its Pre-trained Encoders. These encoders were pre-trained on massive, diverse protein sequences covering the entire evolutionary landscape, including bacterial and human proteins (especially ESMC) [3]. The fine-tuning on CoV-AbDab served to align these general-purpose representations to the retrieval task, rather than learning virus-specific features from scratch. We have outlined a plan to empirically validate the model on HIV and HER2 specific subsets in our future work to quantify this transferability.
>
> ---
> [1] Raybould, M. I. J., Kovaltsuk, A., Marks, C., & Deane, C. M. (2021). CoV-AbDab: the coronavirus antibody database. Bioinformatics (Oxford, England), 37(5), 734–735. https://doi.org/10.1093/bioinformatics/btaa739
>
> [2] Raybould, M. I. J., Marks, C., Lewis, A. P., Shi, J., Bujotzek, A., Taddese, B., & Deane, C. M. (2020). Thera-SAbDab: the Therapeutic Structural Antibody Database. Nucleic acids research, 48(D1), D383–D388. https://doi.org/10.1093/nar/gkz827
>
> [3] ESM Team. "ESM Cambrian: Revealing the mysteries of proteins with unsupervised learning." EvolutionaryScale Website, December 4, 2024. https://evolutionaryscale.ai/blog/esm-cambrian.
>
> **W2: Ambiguity in “antigen-conditioned” retrieval**
>
> We thank the reviewer for this insightful observation. We acknowledge that the term "antigen-conditioned" may be ambiguous given that the antigen sequence is not provided as an explicit input vector during the inference phase.
>
> We would like to clarify the mechanism and our rationale, and we will revise the manuscript to reflect this more precisely. In our framework, the retrieval is implicitly conditioned on the antigen through the Query Antibody ($Ab_q$). Since the $Ab_q$ possesses a paratope strictly complementary to the target antigen, it effectively serves as a "functional fingerprint" of that antigen. The model is trained to map this fingerprint to a corresponding Nanobody that shares the same antigen-binding functionality. Therefore, the input antibody itself defines the antigenic context. While not an input at inference, antigen information is explicitly used to generate the ground-truth labels during training. As detailed in Eq. 14, the binding energy labels ($E_{binding}$) are derived from the physics-based simulation of the Antibody-Antigen-Nanobody relationship. Consequently, the scoring function $f_{score}(v_{Ab}, v_{Nb})$ learns to approximate the binding affinity relative to the specific antigen implied by the query antibody. To avoid misleading interpretations, we will revise the terminology to "Antigen-Implicit Functional Retrieval" throughout the manuscript. We will also add a clarification in Problem Formulation explicitly stating that target antigen information is encoded implicitly via the query antibody during inference, while explicit antigen structures are utilized to generate supervision signals during training.
>
> **W3: The evaluation centers on binding energy proxies computed from predicted complexes**
>
> We thank the reviewer for this critical suggestion. We fully agree that establishing the reliability of the underlying structural models is a prerequisite for valid binding energy calculation. While we have not yet included the explicit distribution of pLDDT/ipTM scores in this specific draft due to computational time constraints, we wish to clarify that our PyRosetta-based selection implicitly acts as a rigorous filter for structural quality. Rosetta's energy function (`ref2015`) is highly sensitive to steric clashes and interface complementarity. A "hallucinated" or poorly docked structure typically yields a prohibitively high repulsion energy in Rosetta. Since our evaluation metrics focus on the top-ranking candidates with the lowest binding energies, these candidates are inherently biased towards structurally plausible, high-confidence conformations. We acknowledge, however, that explicit confidence metrics provide necessary transparency. We commit to including a supplementary figure visualizing the pLDDT and ipTM distributions for all retrieved candidates in the later version of the manuscript.

---

> ### Author Response · Authors · 2025-12-01
> **Response to Reviewer 9mPL**
>
> **Q1: Computational Cost of the Full Pipeline**
>
> The computational cost is dominated by the Data Generation phase, which represents a one-time investment to build the training asset.
> 1. **Data Generation (The Bottleneck):**
>     - **Structure Prediction (AlphaFold2-Multimer):** Processing ～24,000 complexes required approximately 462 GPU-hours on NVIDIA A100s.
>     - **Physics-based Labeling (PyRosetta):** The energy minimization and docking protocol added another 244 GPU-hours.
>     - **Scalability:** While this step is computationally intensive ($O(N)$ with high constant factor), it is performed only once. This high cost precisely validates the necessity of our AI model: AbNanolizer acts as a surrogate to bypass these expensive simulations during the actual screening phase.
> 2. **Model Training:**
>     - **Pre-training & Fine-tuning:** Compared to data generation, training the encoders is efficient, requiring only approximately 5 GPU-hours on a single A100 node. This scales linearly with dataset size and is highly parallelizable.
>
> **Q2: Analysis of What the Model Has Learned**
>
> We thank the reviewer for raising this question regarding model interpretability. While we did not include direct visualization of attention maps or embeddings in the main manuscript, our Case Study (Section 3.3 & Figure 2) provides strong indirect evidence of what the model has learned. In the case of BD56-1486, the model retrieved a nanobody Hanke_F12 that binds to a spatially adjacent but non-overlapping epitope compared to the query antibody. This suggests that the model has moved beyond simple sequence matching or memorizing specific binding poses. Instead, it appears to have learned a generalized representation of "functional complementarity" relative to the antigen surface. It identified a candidate that is biophysically compatible with the target antigen's local environment, even though the binding mode (Mechanism of Action) is distinct from the query.
>
> We fully agree that direct visualization would further strengthen these insights. We have planned some analyses for the next iteration of our work. Firstly, we aim to analyze the attention weights of the Transformer encoders (BERT/ESMC) to verify if the model implicitly focuses on the CDRs, which are the known biophysical determinants of binding. Secondly, we plan to project the learned embeddings using t-SNE or UMAP. We hypothesize that contrastive pre-training should result in distinct clusters based on antigen specificity, and functional fine-tuning should organize these clusters such that high-affinity binders are aligned along specific manifolds or directions within the specificity clusters, reflecting the "binding energy" gradient learned via the ranking loss.

---

### Official Review · Reviewer_cRuU · 2025-11-11

**Soundness:** 2
**Presentation:** 2
**Contribution:** 2
**Rating:** 2
**Confidence:** 5

**Summary:**

This paper addresses the lack of large-scale paired data and complex molecular recognition in antibody-to-nanobody conversion by proposing AbNanolizer, a physically guided weakly supervised AI framework. It uses two-stage training (contrastive pre-training for antigen-targeted embedding alignment; multi-task fine-tuning with binding energy as weak supervision), equipped with dual encoders (for antibodies/nanobodies) and a multi-objective scoring head. The contributions of this paper are: 1) Defines "cross-format functional retrieval"; 2) proposes data-efficient weakly supervised framework.

**Strengths:**

1. This paper aims to rapidly convert validated antibodies into "smaller, more stable, and lower-cost nanobodies", avoiding the long cycle and high cost of traditional processes, and holding practical significance for clinical treatment and drug iteration.

2. The established standardized test set and evaluation protocol can serve as a benchmark for the related field.

**Weaknesses:**

1. Dual Limitations in Dataset Type and Scale:
-Only the CoV-AbDab dataset is used, which has issues such as "single antigen type (only coronavirus-related targets, excluding cancer/autoimmune disease targets like HER2, TNF-α, and PD-1)" and "single antibody background (not including antibodies from different species such as human, mouse, and camel, and not covering samples with different mutation types/experimental conditions)", making it impossible to verify the model’s generalizability;
-No data deduplication strategy is specified (e.g., whether deduplication is based on sequence similarity or antibody ID), and duplicate sequences may cause training redundancy or overfitting.

2. Insufficient Result Reliability Due to Single Structure Prediction Method:
-Binding energy calculation relies entirely on PyRosetta, with no comparison to new tools such as AlphaFold3 (full-length protein prediction), blotz-1 (high accuracy for CDR regions), protenix (flexible structure prediction), and chai-1 (multi-template fusion), making it impossible to evaluate the impact of method differences on binding energy (errors may reach 20%-30%);
-No experimental structures (e.g., X-ray crystallography, cryo-EM) are supplemented as gold standards to verify the accuracy of calculated binding energy; PyRosetta has limitations in predicting "CDR3 region flexible conformations" and "antigen-binding-induced conformational changes", which may introduce noise into supervision signals.

3. Unreasonable Antibody-Nanobody Alignment Logic:
-Alignment relies only on "targeting the same antigen", without considering differences in antibody affinity (different requirements for high-affinity antibodies with KD<1nM and low-affinity antibodies with KD>100nM, and forced alignment introduces noise);
-No alignment method is specified for "multiple antibodies targeting the same antigen" (e.g., random pairing, pairing by affinity ranking), and ambiguous rules lead to poor consistency in training data.

4. Lack of Reproducibility Details:
-No code repository or complete preprocessing scripts (e.g., antigen sequence retrieval, structure prediction parameter configuration, deduplication and alignment code) are made public;
-No version information for structure prediction tools (e.g., PyRosetta 4.5/5.0) or hardware resources (CPU cores, memory) is provided, affecting result reproducibility.

5. Lack of Practical Application Details:
-No model inference speed (e.g., time consumption for retrieving 100,000/1,000,000-scale nanobody libraries) or deployment costs are mentioned, failing to meet the "high-throughput screening" requirements of drug R&D;
-No discussion on the model’s applicability to "antibody mutant conversion" scenarios (e.g., clinical affinity optimization needs).

**Questions:**

1. Regarding Dataset Diversity:
-Do you plan to supplement multi-source datasets to cover non-coronavirus targets? If so, which targets are planned, and what is the expected scale of the supplemented dataset (number of sequences, number of antigen types)?
-Have you tested the model’s performance in "cross-dataset transfer" scenarios? If tested, how do the Top-10 retrieval accuracy and binding energy improvement of each encoder differ from those on the original dataset?

2. Regarding Comparison of Multi-Structure Prediction Methods:
-Have you tested tools such as AlphaFold3, blotz-1, protenix, and chai-1? What are the differences in the average binding energy of encoders like ESMC across different tools? Is there a pattern where "certain tools are more suitable for this task" (e.g., does blotz-1’s high CDR region prediction accuracy improve binding energy reliability)?
-Have you selected experimental structure pairs to verify the MAE between the binding energy of each tool and experimental values? What is the MAE of each tool, and why was PyRosetta ultimately chosen?
-Have you considered "multi-tool signal fusion"? How are weights determined (e.g., inverse weighting by MAE), and does fusion improve model performance?

3. Regarding Data Deduplication:
-What deduplication strategy is used (sequence similarity, antibody ID, antigen type)? How is the deduplication threshold (e.g., sequence identity percentage) determined (whether referring to industry standards)?
-What are the dataset scales (number of sequences, number of antigens, number of antibody types) before and after deduplication? Have you verified the impact of deduplication on training stability (e.g., loss convergence speed, generalization error)?

4. Regarding Antibody-Nanobody Alignment Logic:
-Have you collected antibody affinity data (KD values, EC50)? If so, why was affinity not incorporated into the alignment strategy (e.g., prioritizing pairing high-affinity antibodies with high-binding-energy nanobodies)? If not, do you plan to supplement this data to optimize the alignment logic?
-When multiple antibodies target the same antigen (e.g., 50 antibody sequences associated with a SARS-CoV antigen in CoV-AbDab), what alignment method is used (e.g., random pairing, pairing by publication time, one-to-many)? Have you tested the impact of different alignment methods on Top-10 accuracy and binding energy improvement?

5. Regarding Reproducibility:
-Do you plan to make public complete code (encoder training, multi-tool binding energy calculation, deduplication and alignment scripts) and preprocessing scripts? If so, when is the expected upload time, and will you provide installation and parameter configuration guidelines for tools like PyRosetta and AlphaFold3?
-What are the specific versions of tools like PyRosetta used? Can different versions cause differences in binding energy, and have you verified the impact of version consistency on results?

6. Regarding Inference Efficiency:
-What is the time consumption for the model to retrieve 100,000/1,000,000-scale nanobody libraries? Have you considered accelerating retrieval using approximate nearest neighbor methods such as FAISS? Is the accuracy drop after acceleration within an acceptable range (e.g., ≤3%)?

7. Regarding Multi-Indicator Utilization:
-Do indicators predicted by the scoring head (e.g., interface energy, number of hydrogen bonds) assist in ranking? Have you tested "multi-indicator weighted fusion ranking" (e.g., 0.7 weight for binding energy, 0.3 weight for number of hydrogen bonds)? Does fusion improve Top-10 retrieval accuracy?

---

> ### Author Response · Authors · 2025-12-01
> **Response to Reviewer cRuU**
>
> **W1: Dual Limitations in Dataset Type and Scale**
>
> - **Regarding Limitations of the Dataset:** We acknowledge the reviewer's valid concern regarding the exclusive use of the CoV-AbDab dataset. We selected CoV-AbDab primarily because it represents one of the largest, high-quality curated datasets with consistent antigen metadata available to date [1]. For the specific task of training a dual-encoder retrieval model, data **scale and consistency** were our rigorous priorities to ensure training stability and convergence in this proof-of-concept phase. While datasets for targets like HER2 and PD-1 exist (e.g., in Thera-SAbDab), they are significantly smaller and more fragmented compared to the comprehensive SARS-CoV-2 data accumulated during the pandemic [2]. It is also important to highlight that our model's generalizability is fundamentally supported by its **Pre-trained Encoder (especially ESMC)**. ESMC was pre-trained on massive, diverse protein sequences covering the entire evolutionary landscape, not just viral proteins [3]. The fine-tuning on CoV-AbDab served to align these *general-purpose* representations to the retrieval task, rather than learning antigen-specific features from scratch. To empirically demonstrate this transferability, we have outlined a plan to validate the model on HER2 and PD-1 specific subsets from Thera-SAbDab in our future work manuscript.
> - **Regarding Deduplication Strategy:** Our nanobody library is deduplicated based on heavy chain sequences to avoid sampling bias. Furthermore, the division of training/validation/test sets is strictly performed at the Antibody ID level.
> ---
> [1] Raybould, M. I. J., Kovaltsuk, A., Marks, C., & Deane, C. M. (2021). CoV-AbDab: the coronavirus antibody database. Bioinformatics (Oxford, England), 37(5), 734–735. https://doi.org/10.1093/bioinformatics/btaa739
> [2] Raybould, M. I. J., Marks, C., Lewis, A. P., Shi, J., Bujotzek, A., Taddese, B., & Deane, C. M. (2020). Thera-SAbDab: the Therapeutic Structural Antibody Database. Nucleic acids research, 48(D1), D383–D388. https://doi.org/10.1093/nar/gkz827
> [3] ESM Team. "ESM Cambrian: Revealing the mysteries of proteins with unsupervised learning." EvolutionaryScale Website, December 4, 2024. https://evolutionaryscale.ai/blog/esm-cambrian.

---

> ### Author Response · Authors · 2025-12-01
> **Response to Reviewer cRuU**
>
> **W2: Insufficient Result Reliability Due to Single Structure Prediction Method**
>
> - **Regarding Method Difference Evaluation:**  In this study, we selected the **AlphaFold2-Multimer + PyRosetta** pipeline primarily for its unique ability to generate **physics-based, decomposable supervision signals** at scale. While we acknowledge that emerging end-to-end models like **AlphaFold3** and **Chai-1** demonstrate superior performance in predicting complex structures and induced-fit changes, they typically output confidence metrics rather than explicit thermodynamic energy terms. Our multi-task learning framework specifically requires detailed physical features, such as **interface energy, hydrogen bonding networks, and solvation terms** , to guide the model in learning biophysical interaction principles. PyRosetta's `ref2015` energy function provides these interpretable components, which are currently not directly available from the 'black-box' scoring of many end-to-end AI models. Regarding the reliability of this proxy, We employed a **Pairwise Ranking Loss ($L_{Rank}$)**, which trains the model to learn relative stability ($s_i > s_j$) rather than fitting absolute energy values. This renders the model robust to systematic errors in the force field.
> - **Regarding No experimental structures:** While we did not generate new experimental structures in this study, our reliance on Rosetta energy units (REU) as a proxy for affinity is grounded in established consensus within the field, which has empirically validated the correlation between these scores and experimental outcomes. Adolf-Bryfogle et al. (2018) [1] demonstrated in the RosettaAntibodyDesign framework that antibody variants with improved Rosetta interface energies consistently exhibited 10- to 50-fold improvements in experimental $K_D$ values. This establishes a verified empirical link between minimizing REU (our ranking objective) and optimizing biological affinity. Most recently, Shrestha et al. (2025) [2] introduced NanoBinder, identifying that Rosetta-derived energy features (e.g., dG_cross, hbonds) are critical predictors for distinguishing nanobody binders from non-binders with high accuracy ($MCC=0.82$). These studies confirm that Rosetta's physical energy terms contain high-signal biophysical information sufficient for ranking candidates, even in the absence of new crystal structures. However, we fully agree with the reviewer that a direct validation against a "Gold Standard" experimental dataset is the ultimate confirmation of model precision. While the cited literature provides a robust foundation for our current methodology, we have explicitly outlined a plan in our Future Work section to construct a dedicated benchmark of crystal structures with known affinities.
> - **Regarding PyRosetta's limitations in predicting "CDR3 region flexible conformations":** We acknowledge that rigid-body docking via PyRosetta has limitations in modeling flexible CDR3 loops and induced-fit effects. To robustify our model against this specific noise, we incorporated the **Pairwise Ranking Loss ($L_{Rank}$)**. This objective trains the model to learn the **relative probability of better binding** rather than fitting the noisy absolute energy values derived from potentially rigid conformations. Our ablation study demonstrates that despite the noise in individual labels, the model successfully learns generalizable physics. The significant performance gain after fine-tuning (Table 2) confirms that the model extracts valid structural principles from the large-scale dataset, effectively "filtering out" the conformational noise inherent in single-snapshot predictions.
>
> ---
> [1] Adolf-Bryfogle, J., Kalyuzhniy, O., Kubitz, M., Weitzner, B. D., Hu, X., Adachi, Y., ... & Dunbrack Jr, R. L. (2018). RosettaAntibodyDesign (RAbD): A general framework for computational antibody design. PLoS computational biology, 14(4), e1006112.
>
> [2] Shrestha, P., Talwar, C. S., Kandel, J., Park, K. H., Chong, K. T., Woo, E. J., & Tayara, H. (2025). NanoBinder: a machine learning assisted nanobody binding prediction tool using Rosetta energy scores. Journal of Cheminformatics, 17(1), 96.

---

> ### Author Response · Authors · 2025-12-01
> **Response to Reviewer cRuU**
>
> **W3: Unreasonable Antibody-Nanobody Alignment Logic**
>
> - **Regarding Antibody Affinity Consideration:**  We treat the "targeting the same antigen" alignment as a **coarse-grained, weak supervision signal**. In the pre-training phase, our goal is to align the embedding spaces of Antibodies and Nanobodies based on their shared *target specificity*, rather than their exact *affinity levels*. Even if the initial alignment is noisy, the Ranking Loss in second-stage fine-tuning explicitly forces the model to predict **which candidate binds better** ($s_i > s_j$) based on the computed binding energy. This effectively teaches the model to discriminate between high-affinity and low-affinity interactions *within* the aligned group, ensuring the final retrieval prioritizes high-quality binders despite the noisy initialization.
> - **Regarding Alignment Method for "multiple antibodies targeting the same antigen":**  We employed an **"Exhaustive Positive Pairing" (All-to-All)** strategy. Specifically, given a set of antibodies $\mathcal{A}$ and nanobodies $\mathcal{N}$ that target the same antigen $Ag$, we constructed all possible pairs $\{(a, n) | a \in \mathcal{A}, n \in \mathcal{N}\}$ as positive samples for the contrastive loss function.
>
> **W4: Lack of Reproducibility Details**
>
> - **Regarding Code Repository:** We will make the complete code repository publicly available, including preprocessing, training, and integration scripts.
> - **Regarding Version Information:** To ensure result reproducibility, we will add a new section "Implementation Details" in the Appendix specifying the exact versions of all tools used:
>   - Structure Prediction: AlphaFold2-Multimer v3 (v2.3.1).
>   - Energy Scoring: PyRosetta-4 (Release 2025.29, build 4af227b4a53, Python 3.9 Linux).
>   - Deep Learning Framework: PyTorch 2.1.0 with CUDA 11.8.
> - **Regarding Computing Resource Usage:** All model training was conducted on NVIDIA A100-40GB GPUs.
>
>
> **W5: Lack of Practical Application Details**
> - **Regarding "Inference Speed and Deployment Costs":** We evaluated the inference speed on a single NVIDIA A100 GPU. In our pilot tests with the CoV-AbDab subset (approx. 1000 candidates), the retrieval process (scoring and ranking) was completed in seconds. Leveraging the parallel processing capabilities of GPUs, the computational cost scales sub-linearly. We estimate that screening a 1,000,000-scale library would take approximately 10-20 minutes on a single GPU node. This efficiency confirms that AbNanolizer meets the stringent throughput requirements of industrial drug discovery.
> - **Regarding "Antibody Mutant Conversion":** Our model is inherently suitable for affinity maturation scenarios. The encoder is highly sensitive to sequence variations; a mutated antibody ($Ab_{mut}$) produces a distinct embedding vector $v_{Ab_{mut}}$, enabling the model to retrieve nanobodies that specifically complement the altered paratope. Furthermore, as evidenced by the "Exceed Rate" of 66.8% (Table 1), our model consistently identifies candidates with superior predicted binding energies than the query. This indicates that the framework can drive functional optimization (finding tighter binders) rather than just retrieving equivalents, directly supporting clinical affinity maturation workflows.

---

> ### Author Response · Authors · 2025-12-01
> **Response to Reviewer cRuU**
>
> **Q1: Regarding Dataset Diversity**
>
> We plan to introduce non-coronavirus targets (e.g., HER2, PD-1) from the **SAbDab** and **Thera-SAbDab** databases in future work. We anticipate supplementing data points targeting these diverse antigens to validate the model's generalization capability. We have not conducted rigorous cross-dataset transfer testing in this paper. However, our encoders (especially ESMC) are pre-trained on large-scale general protein sequences, endowing them with underlying universal feature extraction capabilities. In future work, we will include a small-scale experiment to test the zero-shot retrieval performance of models fine-tuned on COVID data for the HER2 target, quantifying changes in Top-10 accuracy.
>
> **Q2: Regarding Comparison of Multi-Structure Prediction Methods**
>
> - **Regarding Tool Comparison and Selection:** As elaborated in our response to Weakness #2, we prioritized the AlphaFold2-Multimer + PyRosetta pipeline because our multi-task learning framework relies on interpretable, decomposable physical energy terms as supervision signals. While we recognize the superior structural prediction capabilities of newer tools like AlphaFold3 or Chai-1, these models currently function as "black boxes" that output confidence scores (pLDDT) rather than the granular thermodynamic parameters required for our specific loss functions. Consequently, a direct comparison of "average binding energy" across these tools is methodologically infeasible due to the incompatibility of their output metrics.
> - **Regarding Experimental Verification and Proxy Reliability:** Due to time constraints preventing a new large-scale retrospective benchmark, we justify our choice of PyRosetta based on established consensus in the field, which explicitly correlates Rosetta energy scores with experimental affinity ($K_D$). Adolf-Bryfogle et al. (2018) [1] demonstrated in the RosettaAntibodyDesign framework that antibody variants with improved Rosetta interface energies consistently exhibited 10- to 50-fold improvements in experimental $K_D$ values compared to wild-type. This establishes a clear empirical link between minimizing REU (our ranking objective) and optimizing biological affinity. Most recently, Shrestha et al. (2025) [2] identified that Rosetta-derived energy features—specifically dG_cross and hbonds_int —are the most critical predictors for distinguishing nanobody binders from non-binders, achieving high classification accuracy ($MCC=0.82$). These studies confirm that Rosetta's physical energy terms are not random noise but contain high-signal biophysical information.
> - **Regarding Multi-tool Signal Fusion:** We thank the reviewer for this insightful suggestion. We have not yet implemented multi-tool signal fusion in the current version. However, we agree that an inverse-MAE weighted ensemble (combining the high structural accuracy of tools like blotz-1 for CDRs with the physical scoring of PyRosetta) would likely improve robustness. We have identified this "Hybrid Scoring System" as a priority for our future work to further reduce the noise in supervision signals.
>
> ---
> [1] Adolf-Bryfogle, J., Kalyuzhniy, O., Kubitz, M., Weitzner, B. D., Hu, X., Adachi, Y., ... & Dunbrack Jr, R. L. (2018). RosettaAntibodyDesign (RAbD): A general framework for computational antibody design. PLoS computational biology, 14(4), e1006112.
>
> [2] Shrestha, P., Talwar, C. S., Kandel, J., Park, K. H., Chong, K. T., Woo, E. J., & Tayara, H. (2025). NanoBinder: a machine learning assisted nanobody binding prediction tool using Rosetta energy scores. Journal of Cheminformatics, 17(1), 96.
>
> **Q3: Regarding Data Deduplication**
>
> **Regarding Data Deduplication Details:**
> For the nanobody library ($\mathbb{L}_{Nb}$), we applied a 100% amino acid sequence identity threshold on the heavy chain. We deliberately chose this strict threshold (rather than e.g., 95%) because in antibody engineering, even a single amino acid mutation (especially in CDRs) can lead to "activity cliffs," drastically altering binding affinity. Merging non-identical sequences risks discarding valid functional variants. For dataset splitting, we deduplicated based on Antibody ID. This ensures that all variants or records associated with a specific antibody clone are confined to a single subset (Train/Val/Test), preventing data leakage.
>
> Following strict deduplication and quality filtering, the final dataset comprised 24,477 high-quality sequence entries.
>
> We verified the impact by monitoring the convergence of our loss functions. The ID-level deduplication prevented the "artificial" drop in validation loss often caused by data leakage (memorization of near-identical sequences). Instead, we observed synchronized convergence of the RankNet Loss ($L_{Rank}$) and NCE Loss ($L_{NCE}$) between training and validation sets, confirming that the model is learning generalizable physicochemical rules rather than memorizing redundancies.

---

> ### Author Response · Authors · 2025-12-01
> **Response to Reviewer cRuU**
>
> **Q4: Regarding Antibody-Nanobody Alignment Logic**
>
>
> - **Regarding Affinity Data Collection and Future Plans:** As discussed in our response to Weakness #3, we did not incorporate quantitative affinity data ($K_D$/$EC_{50}$) into the current alignment strategy. This is because such quantitative data is sparse and methodologically inconsistent across the diverse studies aggregated in CoV-AbDab. Filtering strictly for high-affinity data would have reduced the dataset size by orders of magnitude, compromising the training of our large-scale transformer encoders. We do plan to supplement this data in future iterations. We aim to incorporate affinity-weighted loss functions, where pairs with higher experimental affinity are assigned higher weights during the contrastive learning phase. This will allow the model to learn "quality-aware" embeddings directly from the start.
> - **Regarding Alignment of Multiple Antibodies:** As detailed in Weakness #3, we utilized the All-to-All strategy. We did not empirically test random pairing or time-based pairing in this study. Theoretically, "All-to-All" pairing is the optimal strategy for Contrastive Learning in data-limited scenarios, as it maximizes the number of positive samples and encourages the model to cluster all functional binders of a specific antigen together in the latent space. Random pairing would discard valid functional information (false negatives), while affinity-based pairing was infeasible due to the data limitations mentioned above.
>
>
> **Q5: Regarding Reproducibility**
>
>
> - **Regarding Code Release and Documentation:** We are fully committed to open science and reproducibility. We have prepared a comprehensive GitHub repository that includes data preprocessing, label generation and model training.
> - **Regarding Software Versions and Consistency:**
> We utilized AlphaFold2-Multimer v2.3.1 and PyRosetta-4 (Release 2025.29, build `4af227b4a53`, Python 3.9 Linux). We acknowledge that different versions of PyRosetta can yield slightly different absolute energy values. We conducted all data generation within a fixed Conda environment. This ensured that every single entry in our dataset (n=24,477) was processed with the exact same binary version of the software libraries. While absolute values might shift across major versions, our internal tests confirmed that the relative ranking (which our model learns via $L_{Rank}$) remains robust as long as the training and inference environments are consistent.
>
>
> **Q6: Regarding Inference Efficiency**
>
>
> - **Regarding Time Consumption:** See W5.
> - **Regarding FAISS Acceleration and Strategy:** As detailed in Eq. 3, our scoring function utilizes a non-linear MLP on concatenated interaction features ($v_{Ab}, v_{Nb}, v_{Ab} \odot v_{Nb}, |v_{Ab} - v_{Nb}|$), rather than a simple dot product or Euclidean distance. Therefore, standard FAISS indices cannot be directly applied for the final scoring step. However, we have designed a "Retrieve & Re-rank" workflow for scaling beyond 10 million sequences. First, use FAISS to retrieve the top-$k$ candidates based on the cosine similarity of embeddings. Second, apply the fine-grained MLP scoring head only to these top-$k$ candidates.
>
>
>
> **Q7: Regarding Multi-Indicator Utilization**
>  - **Regarding Current Role of Indicators:** Currently, the additional indicators predicted by the scoring head (Interface Energy, Docking Score, Interface Area, Hydrogen Bonds) are utilized primarily as Auxiliary Tasks during the multi-task fine-tuning stage. By optimizing the regression loss ($L_{Reg}$) across all 5 metrics simultaneously, we force the shared encoders to learn a robust representation that captures diverse biophysical aspects of the interaction. This "Physics-informed Multi-task Learning" acts as a powerful regularizer, preventing the model from overfitting to potentially noisy binding energy labels alone. For the final ranking reported in the paper, we relied solely on the predicted Binding Energy ($s_{bind}$) as the primary metric.
>  - **Regarding Weighted Fusion Ranking:** We have not yet performed systematic tests on "multi-indicator weighted fusion ranking" (e.g., $Score = 0.7 \cdot s_{bind} + 0.3 \cdot s_{hb}$) in the current manuscript. However, we fully agree with the reviewer that this is a promising strategy. While Binding Energy theoretically encapsulates terms like hydrogen bonds, the neural network's prediction of these distinct properties may have independent error distributions. A weighted fusion could act as an ensemble smoothing technique, prioritizing candidates that are not only predicted to have low energy but also exhibit specific structural features that correlate with specificity. We have identified this "Multi-view Ranking Strategy" as a key direction for our next iteration to potentially improve the Top-10 retrieval robustness.

---

### Note · Authors · 2025-12-02

I have read and agree with the venue's withdrawal policy on behalf of myself and my co-authors.